# BYOC: Personalized Few-Shot Classification with Co-Authored Class Descriptions

**Arth Bohra**[*]
University of California Berkeley
Berkeley, CA, USA
arthbohra@berkeley.edu

**Govert Verkes**
**Artem Harutyunyan**
**Pascal Weinberger**
**Giovanni Campagna**
Bardeen, Inc.
San Francisco, CA, USA
{govert,artem,pascal,giovanni}@bardeen.ai

## Abstract

Text classification is a well-studied and versatile building block for many NLP applications. Yet, existing approaches require either large annotated corpora to train a model with or, when using large language models as a base, require carefully crafting the prompt as well as using a long context that can fit many examples. As a result, it is not possible for end-users to build classifiers for themselves.

To address this issue, we propose a novel approach to few-shot text classification using an LLM. Rather than few-shot examples, the LLM is prompted with descriptions of the salient features of each class. These descriptions are coauthored by the user and the LLM interactively: while the user annotates each few-shot example, the LLM asks relevant questions that the user answers. Examples, questions, and answers are summarized to form the classification prompt.

Our experiments show that our approach yields high accuracy classifiers, within 79% of the performance of models trained with significantly larger datasets while using only 1% of their training sets. Additionally, in a study with 30 participants, we show that end-users are able to build classifiers to suit their specific needs. The personalized classifiers show an average accuracy of 90%, which is 15% higher than the state-of-the-art approach.

## 1 Introduction

Text classification – the task of mapping a sentence or document to one class of a predefined set – is a well-studied, fundamental building block in natural language processing (Maron, 1961; Li et al., 2022), with applications such as textual entailment (Dagan et al., 2005; Bowman et al., 2016), sentiment analysis (Wang and Manning, 2012; Socher et al., 2013), topic classification (Hingmire et al., 2013), intent and dialog act classification (Godfrey et al.,

**1. User sets context and initial class descriptions of their task**

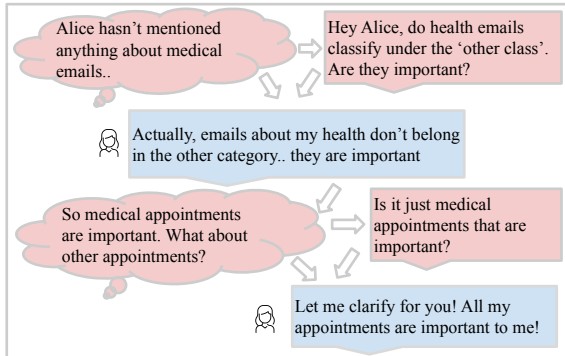

**2. User annotates a selected example**

Dear Alice,
Here is an update on your health. Your medical tests indicate you need to purchase the following treatment: turmeric. Also, feel free to schedule an appointment next week!
- Dr. Nate

**2a. Generating and answering questions about the example**

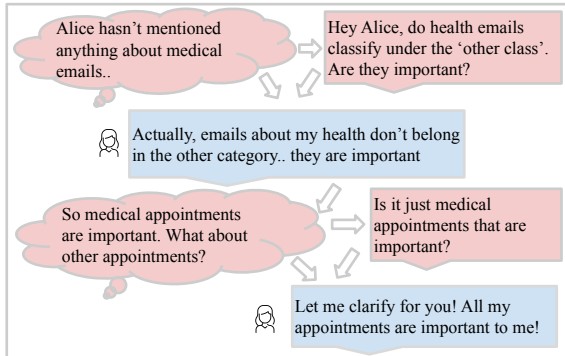

**2b. Validating the model**

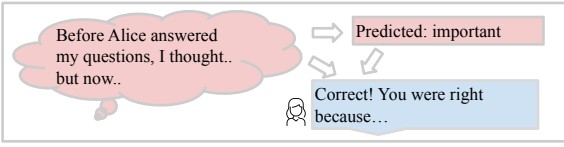

**2c. Refining class descriptions**

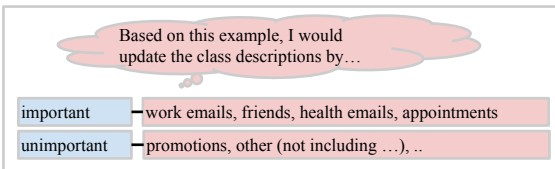

**3. Using the Classifier**

Dear Alice,
Your biology degree requires you to get at least four credits of mathematics. Join a 'Math Call' with your major advisor to figure out what you can take!
- University

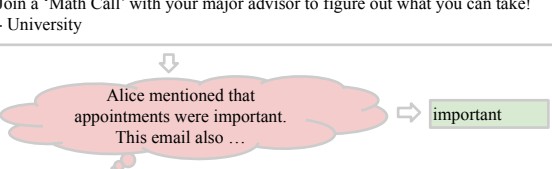

Figure 1: Overview of how a user can construct and use a classifier with BYOC.

---

[*] Work done while at Bardeen

1992; Reithinger and Klesen, 1997; Tur et al., 2010; Coucke et al., 2018; Xu et al., 2021), and more.

Because a classification task targets a fixed set of classes, each of these many applications of classification must be built separately. Furthermore, classes are often highly domain-specific, and in certain scenarios must be personalized to each user. Consider the example of classifying an email as "Important" or "Not Important". Many users would consider family emails to be important, but some might decide to only prioritize work emails, or only emails that require immediate action such as a reply. As a more extreme example, most users would consider coupons to be junk mail, but some might actively search for savings.

There exist a plethora of use cases where personalized classification is necessary:

- Classifying product or business reviews as useful according to someone's preferences.

- Classifying user support requests as being pertinent to a particular product component, so they can be routed to the relevant team.

- Classifying an academic paper as likely relevant or not to someone's research.

In all of these contexts, annotating a large dataset and then training a domain-specific model is intractable. It is too expensive and time-consuming for individual end-users to label the hundreds of examples needed to train a high-quality classifier, and even more so if the criteria change frequently, because each time the training set would need to be rebuilt from scratch. Hence, we need a technique that can work well in a *few-shot* regime, where the user only labels a limited number of examples.

To overcome this challenge, we propose to leverage Large Language Models (LLMs), which have been shown to perform very well on a variety of classification tasks (Radford et al., 2019; Raffel et al., 2020; Brown et al., 2020; Gao et al., 2021).

Prior work using LLMs for few-shot text classification proposed using *task demonstrations* consisting of examples and their labels, and proposed to include as many as possible to fit in the context length (Brown et al., 2020; Gao et al., 2021). There are two issues with this approach: (1) the context length is limited, and with longer text only few examples can fit, and (2) the prompt length can be significantly larger than the size of a single input, increasing the computational cost.

At the same time, for many of these tasks, the mapping from text to classes can be expressed in relatively simple rules. LLMs based on instruction tuning (Ouyang et al., 2022) can classify according to instructions expressed as text. The challenge then becomes: how can we help end-users write meaningful and sufficient instructions? End-users are not prompt engineers, and struggle to write good instructions. In our experiments we found that users often forget about relevant situations or conditions, leading to poor quality classification.

We also observe that LLMs can be prompted to elicit information and ask clarifying questions. Hence, we propose an interactive approach to help users write high-quality classification prompts, which we call *Build Your Own Classifier* (BYOC):

1. The user provides the purpose of the classifier and the list of classes.

2. They annotate a small amount of data. For each example, the LLM generates questions that would be helpful to classify the example. The user answers the questions, then selects the correct class for the example and explains why they selected that class.

3. The examples, questions, answers, labels, and explanations are summarized into *class descriptions*: succinct textual representations of the classification criteria and relevant aspects of the training data.

4. At inference time, the LLM is prompted with only the generated class descriptions.

Our experiments show that with only a small amount of additional work for each training example, BYOC yields higher accuracy than including training data in the LLM prompt. Indeed, we can improve the accuracy of few-shot text classification to the point where we can be within 79% of fine-tuning a dedicated model on a full dataset.

Our method is also suitable for personalization. In an experiment conducted directly with end users, our approach improves over manually provided class descriptions by 23%, and over the state of the art by 15%, while consuming 37% fewer tokens at inference time.

## 1.1 Contributions

The contributions of this paper are as follows:

- A novel approach to few-shot text classification using class descriptions in lieu of task demonstrations. This approach is more computationally efficient due to a shorter prompt.

- A method, called BYOC, to interactively construct class descriptions for classification, by prompting the LLM to ask relevant questions to the user. Ours is the first approach to use LLM-generated questions to help users craft high quality prompts.

- Experimental results on the Web of Science dataset (Kowsari et al., 2017) show that a classifier built with BYOC is 9% better than the few-shot state of the art, and reaches 79% of the accuracy of the state-of-the-art model trained on a full dataset, using only 1% of their training set.

- BYOC enables non-experts to build classifiers with high-accuracy, with an average accuracy of 90%, 15% higher than the equivalent few-shot baseline and 23% higher than asking users to write prompts directly. Additionally, the classifiers built with BYOC outperform a model trained for all users at once, which underscores the need for personalization.

- We built an end-to-end version of BYOC within the Bardeen web automation platform[1]. In a study with 30 participants, we find that users find our approach interpretable, and 80% would consider using it.

## 2 Related Work

**Few-Shot Text Classification**   Text classification using few-shot training sets has been studied before. Previous methods proposed dedicated model architectures, based on prototype networks (Snell et al., 2017; Sun et al., 2019) or contrastive representation learning (Yan et al., 2018; Chen et al., 2022). These methods cannot take advantage of large language models, especially API-based LLMs that are only available as a text-based black box.

With LLMs, text classification is usually either zero-shot through providing instructions and class names (Radford et al., 2019; Raffel et al., 2020; Ouyang et al., 2022; Sun et al., 2023), through transfer learning from existing tasks (Wang et al., 2021), or few-shot by task demonstrations (Brown

[1] https://bardeen.ai

et al., 2020; Gao et al., 2021), optionally with explanations (Lampinen et al., 2022). Task demonstrations are computationally expensive and limited by the language model context window size. Additionally, it is unclear whether the model can learn from demonstrations effectively (Min et al., 2022). Finally, it is known that end-users struggle to write high accuracy classification prompts (Reynolds and McDonell, 2021; Jiang et al., 2022).

**Prompt Construction**   Concurrent to this work, Pryzant et al. (2023) proposed automatically optimizing the prompt given few-shot data, using a search algorithm and LLM-generated feedback. Wang et al. (2023) proposed summarizing few-shot examples to obtain succinct prompts. Neither of those methods incorporates user feedback to disambiguate challenging examples.

Kaneko et al. (2023) also proposed leveraging question generation for classification. They propose to use interactivity at inference time; our goal instead is to be able to classify non-interactively, and only use questions at training time.

Finally, we note that natural language feedback was also proposed for explainability (Marasović et al., 2021; Yordanov et al., 2022; Wiegreffe et al., 2022), but those approaches did not lead to an improvement in accuracy.

**Data Augmentation**   A different line of work proposed to augment the training data with automatically generated or automatically labeled data.

In particular, one line of work observed that often a large unlabeled corpus is available, and proposed using programmatic rules to label it in a weakly supervised fashion (Ratner et al., 2016, 2017). Later work then proposed learning those rules from natural language explanations (Hancock et al., 2018) or extracting those rules from a pretrained language model (Schick and Schütze, 2021; Pryzant et al., 2022). Weak supervision is not practical for end-user personalized classification, because crafting the rules requires a high level of expertise. Refining the rules also requires a large labeled validation set, which is often not available.

A different line of work proposed to use LLMs to synthesize training data, which is then used to train a smaller classifier (Yoo et al., 2021; Ye et al., 2022; Gao et al., 2023; Møller et al., 2023). As with using LLMs directly for classification, the challenge is to instruct the LLM so the resulting training data is personalized, and not a statistical average across

many users. At the same time, our approaches are complementary, and synthesis could be used in the future to distill the LLM to a more computationally efficient model.

## 3 Overview of BYOC

In this section, we present the process through which a user can build a personalized classifier using BYOC. We do so by way of an example: Alice wants to build an email classifier to categorize emails as "Important" and "Unimportant". The overall flow is summarized in Fig. 1.

### 3.1 High Level Planning

As the first step towards building a classifier, Alice provides a high-level plan of what she wishes to classify, which provides the model with an initial understanding of her objective. First, Alice writes a short purpose statement of what she would like to accomplish: "I want to separate spam from my important emails". Then, she specifies the list of classes she wants to classify into – two classes, in this example. For each class, Alice can optionally write an initial description of what kind of text fits in each class. For example, she can say that work emails are important, and promotions are not. The descriptions play an important role in BYOC and will be iteratively improved in later steps.

### 3.2 Interactive Training

Without training, the model would have a limited understanding of Alice's specific needs for classification. Instead, Alice interactively trains the model with a small amount of data. The goal is two-fold: for Alice to understand the model's decision-making process to refine the prompt, and for the model to learn about difficult examples and specific criteria relevant to the task.

**Training Data**  First, the user selects a source of data to use for training. In our example, Alice selects her email account, and BYOC uses her credentials to fetch and show her an email.

**Interactive QA**  Below the email, BYOC presents Alice with a question, automatically generated from the text of the email. The goal of the question is to clarify the meaning of the text in the context of classification. Along with the question, BYOC displays the model's reasoning for asking the question, whether it is to clarify, broaden, or

confirm what is stated in the class descriptions. Alice answers the question, addressing any confusion the model may have. Then Alice is presented with another question related to the same email, either a follow-up question or a new question altogether. Alice answers again, and the process repeats up to a configurable number of questions for each email.

**Prediction and Labeling**  After Alice answers all questions, the model uses the responses to predict the class of the current email. BYOC displays the prediction and its reasoning, which enables Alice to assess the model's decision-making process. Alice then enters the correct classification of the email, and also corrects the model's reasoning, if necessary.

**Prompt Refinement**  Given the text, questions, answers, classification, and explanation for the current example, BYOC automatically updates the class descriptions, adding or removing any information to improve classification accuracy. Alice then repeats this process on a new email, until she chooses to stop adding new data to the training set.

After annotating all training examples, Alice can review the final refined class descriptions. She can optionally edit the class descriptions to her liking, making any corrections or adding any information that might be missing.

Finally, Alice assigns a name to the classifier and saves it so she can use it later.

### 3.3 Using the Classifier

After Alice creates the classifier, she can use it as an action in Bardeen. For instance, she builds a program that scans her inbox, applies the constructed classifier to each email, and if the email is classified as "not important", it is marked as read and archived. Alternatively, she can use a program that triggers any new email and sends her a text message when the email is classified as important. The same classifier, once built, can be used in different programs. Moreover, she can share it with others, like family members with similar email needs.

## 4 Prompting for Personalized Classification

In this section, we first present the formal definition of a classifier model that uses class descriptions, and then present a method to interactively compute those class descriptions.

## 4.1 Problem Statement

Given a sample $(x, y)$, where $x$ is the input text to be classified and $y \in C$ is the class, our goal is to learn a function $f$ parameterized on $n_C$ and $d_C$

$$f(x; n_C, d_C) = \hat{y}$$

such that, with high probability $\hat{y} = y$, that is, the predicted class is equal to the true class. $n_C$ is the set of user-defined names of all classes $c \in C$, and $d_C$ is the set of all *class descriptions* $d_c$.

We learn $d_C$ from the user-provided classifier purpose $p$, the class names $n_C$, and a small set of labeled samples $(x_i, y_i) \in \mathcal{D}$:

$$d_C = \text{BYOC}(p, n_C, \mathcal{D})$$

The function BYOC that computes the class descriptions is an interactive function: in addition to the given inputs, it has access to an oracle (the user) who can answer any question posed to them in natural language.

## 4.2 Computing Class Descriptions

BYOC proceeds iteratively for each sample $(x_i, y_i)$, updating $d_{C,i}$ at each step. The initial $d_{C,0}$ is provided by the user.

At each step, we ask the user questions in succession. The set of question and answers $Q_i$ is initially empty, and it gets updated with each question $q_{i,t}$ and answer $a_{i,t}$ ($1 \leq t \leq M$):

$$
\begin{aligned}
Q_{i,0} &= \emptyset \\
q_{i,t} &= \text{GenQuestion}(x, d_{C,i-1}, Q_{i,t-1}) \\
a_{i,t} &= \text{User}(q_{i,t}) \\
Q_{i,t} &= Q_{i,t-1} \cup \{(q_{i,t}, a_{i,t})\}
\end{aligned}
$$

"User" denotes the user answering the question.

After collecting $M$ questions, where $M$ is a hyperparameter, BYOC computes:

$$
\begin{aligned}
\hat{y}_i, \hat{e}_i &= \text{InteractivePredict}(x_i, p, n_C, d_{C,i-1}, Q_{i,M}) \\
d_{c,i} &= \text{Update}(d_{C,i-1}, c, x_i, Q_{i,M}, \hat{y}_i, y_i, e_i)
\end{aligned}
$$

where $\hat{e}_i$ is a model-provided explanation for why the sample has the given class, which we obtain through self-reflection (Shinn et al., 2023), and $e_i$ is the user's own rewriting of $\hat{e}_i$.

The above procedure makes use of the following functions implemented with LLMs:

- *GenQuestion*: The model generates a question to clarify the specification of the classification task. We prompt the model to generate questions that refine or broaden the scope of the current class descriptions, and improve the LLM's general understanding of the classification task. In particular, we prompt the model to produce questions that uniquely depend on the user, that are not answerable based on the given text, that are not too similar across examples, and that are not too specific to a given example. The model is also provided the previous questions and answers, and instructed to avoid producing redundant questions. The prompt was refined based on our initial experiments, in order to improve the informativeness of questios.

- *InteractivePredict*: The model attempts to make its own classification of the text based on the current class descriptions and the questions answered by the user for this sample. Access to the answers from the user, which is not available for regular prediction, improves the accuracy and the quality of the reflection obtained from the model.

- *Update*: The model computes a new class description for a given class, given the current class descriptions, the input, questions and answers, model- and user-generated label, and user explanation. The model is prompted to incorporate the information from the new answers, without discarding relevant information from the existing descriptions.

  The model sees both the correct and predicted labels. It is important to include both because it allows the model to update both the class description of the correct class and the description of the incorrect one that was initially predicted.

These functions are implemented with individual calls to a black box LLM. All functions incorporate chain of thought prompting (Wei et al., 2022). Detailed prompts are included in Appendix B.

## 4.3 Classifying With Class Descriptions

After obtaining the final class descriptions $d_C$, we compute a classification on a new sample $x$ as:

$$f(x) = \text{Predict}(x, p, n_C, d_C)$$

The function Predict concatenates a fixed preamble, the class names, descriptions, and classifier purpose, and the input $x$ as a prompt to the LLM,

and then returns the class $c \in C$ whose name $n_c$ corresponds to the output of the LLM. If no class corresponds to the output of the LLM, an error is returned; note that this rarely happens in practice. As before, we apply chain of thought to the LLM.

## 4.4 Handling Long Inputs

For certain problems a single input $x$ can be too long to fit in the LLM prompt. To account for this, we propose to replace $x$ with a summarized version, up to a threshold size $K$ in the number of tokens:

$$\hat{x} = \begin{cases} \text{Summarize}(x, n_C, p) & |x| > K \\ x & \text{otherwise} \end{cases}$$

where $n_C$ are the class names and $p$ is the classifier purpose. The threshold $K$ strikes a balance between computation cost, space in the context for the class descriptions, and space for the input.

We propose a novel summarization approach specifically optimized for classification. First, we split the input into chunks $x_j$ of equal size, approximately respecting paragraph and word boundaries. We then compute the summary of each chunk $s_j$, given the previous summary and the overall purpose of the classifier $p$:

$$s_1 = \text{SummarizeChunk}(p, x_1)$$
$$s_j = \text{SummarizeChunk}(s_{j-1}, p, x_j)$$

SummarizeChunk is implemented by a call to the LLM. The prompt is included in the appendix.

## 5 Evaluation

In this section, we evaluate the quality of our proposed approach. Our main experiment is a personalized classification use-case where we attempt to classify emails according to the user's provided criteria. For comparison, we also evaluate against an existing dataset for the classification of long text.

## 5.1 Experimental Setup

We compare the following approaches:

- *Zero-Shot*: the model is prompted with class names and class descriptions provided by users directly, followed only by the input to classify. Inputs longer than the LLM context window are truncated.

- + *Summary*: we use the same prompt as in Zero-Shot, but additionally inputs longer than the LLM context window are summarized as described in Section 4.4.

- + *Few-Shot*: in addition to the Zero-Shot prompt, the prompt includes few-shot training examples, up to the token limit. Each few-shot example consists of input and label. All inputs are summarized, if necessary.

- + *Explanation*: as in the previous model, but additionally the explanation from the user is included for each few-shot example. This is similar to the approach used by Lampinen et al. (2022).

- + *QA*: as in the previous model, but each few-shot example includes both the explanation and all questions generated by BYOC for that example and the answers from the user.

- *BYOC* (our approach): the model is prompted with class names and class descriptions obtained as described in Section 4.2.

We use the GPT-3.5-Turbo model as the LLM while constructing the classifier interactively, and the GPT4 model (OpenAI, 2023) as the LLM during evaluation on the validation and test set. We used GPT-3.5-Turbo in the interactive part for reasons of speed, where it can be a factor for usability. We use a temperature of 0 for classification to maximize reproducibility, whereas we use a temperature of 0.3 to generate questions, in order to make them more varied and creative.

## 5.2 Personalized Classification

Our main goal is to support personalized classification. To evaluate BYOC ability to do so, we perform a user study, in which we ask a group of users to try and use BYOC to classify their emails. The study is both a way to collect real data to evaluate BYOC as a model, and a way to study how well end-users can interact with BYOC.

**Experimental Setting** We have built an end-to-end prototype of BYOC with a graphical user-interface, based on the Bardeen web browser automation platform. We recruit 30 English-speaking users, 16 women and 14 men, to participate in our study. 10% of the users are above the age of 50 years old, 20% are between the ages of 30 and 50, and 70% are under the age of 30. 13 users are in a technical field, 9 are pursuing business, 6 are studying medicine, and 2 are in the arts.

We ask each user to build a personal classifier to categorize their emails as important or unimportant. We use 30 emails from each user as the data source,

|            | Train   | Dev     | Test    |
|------------|---------|---------|---------|
| # Examples | 300     | 300     | 300     |
| # Tokens   | 473,155 | 586,165 | 668,438 |

Table 1: Size of the BYOC email classification dataset.

| | # Tokens | | |
|------------|-------|-------|----------|
| Approach | Build | Run | Accuracy |
| Zero-Shot | **0** | **1,628** | 66.7% |
| + Summary | **0** | 3,447 | 71.0% |
| + Few-Shot | **0** | 5,535 | 74.7% |
| + Explanation | **0** | 5,590 | 74.7% |
| + QA | 93,282 | 5,818 | 76.7% |
| BYOC | 170,411 | 3,681 | **90.0%** |
| Fine-tuned | n/a | n/a | 78.3% |

Table 2: Test-set accuracy and token counts of our proposed approach against the different baselines for personalized email classification. Fine-tuned refers to a single model trained on all training data simultaneously, while other models are trained for each user separately.

out of their most recent 100. For HTML emails, we remove HTML tags and only use plain text.

10 of those emails are used in training: the user annotates them during the interactive training flow (Section 3.2), together with the answers to the questions generated by BYOC and the explanation for the classification. Each user answers 3 questions for each training email. The remaining 20 emails are used for validation and testing; users annotate them at the end after constructing the classifier with BYOC. 10 of emails for each user are used for validation and 10 for testing.

**Dataset** We acquired a dataset of 900 emails from the 30 users participating in our study. Due to the nature of most emails in an inbox being spam or unimportant to users, 36% of the emails are classified as important, with the rest being unimportant. Statistics of the dataset are shown in Table 1.

**Overall Accuracy** Table 2 shows the results of our experiment. The table shows the accuracy of the classifier on the test sets, as the percentage of examples classified correctly across all users. The table also shows the number of prompt tokens necessary to construct the classifier on average across users ("# Tokens Build"), and to perform the classification on average across emails ("# Tokens Run").

BYOC improves over the state of the art of classification with user explanations (Lampinen et al., 2022) by more than 15%. The zero-shot approach achieves a respectable accuracy of 67%. Adding summarization improves by 4% by avoiding sudden truncation. Few-shot examples further improve accuracy by 3%, but we find no measurable improvement by adding explanations. Asking questions to the user is marginally effective.

Overall, our results are two-fold. First, it is clear that merely asking users to provide class descriptions is not sufficient, and training data is necessary. Second, our results show that the manner in which the training data is annotated and is provided to the model is critical to achieve good accuracy. Not only the questions answered by the user make a difference compared to the existing approach of explanations, the biggest improvement appears when the information collected during the annotation stage is summarized into co-authored class descriptions. BYOC is the first method that enables the model to extract key insights from every example in the training set, and discard the large amount of otherwise irrelevant information.

BYOC is also more token-efficient when compared to previous approaches. Even though it takes on average 170,411 tokens to build an email classifier for a user, this cost is fixed. BYOC uses 1,909 fewer tokens for every example compared to few-shot examples, and is comparable in cost to zero-shot with summarization. In summary, besides the initial fixed cost when building the classifier, BYOC is more accurate and cost-effective than the state of the art.

**Importance of Personalization** In many scenarios, it would be impractical for each users to acquire enough training data to fine-tune a personalized model. Hence, we also wish to evaluate the tradeoff between a few-shot personalized model, compared to a fine-tuned model that uses more training data across different users. To do so, we train single fine-tuned model using all 300 emails in the training set at once. We use a distilled BERT model (Sanh et al., 2019) with a classifier head connected to the embedding of the first token.

Results for this experiment are shown in Table 2, in the "Fine-Tuned" line. The fine-tuned model achieves higher accuracy than LLM baselines, but BYOC still improves the accuracy by 13%. The increase in accuracy is likely due to everyone's differences in perception of what makes an important or unimportant email. BYOC is more accurate

because it enables the LLM to receive personal information about each user.

**User Evaluation** At the end of the study, we ask each user to also complete a short survey to gather feedback on their interaction with BYOC.

Overall, on average on a scale of 1 to 10, users rate their experience with BYOC an 8.6. When asked about whether they would actually take the time to build a classifier, on a scale of 1 to 10, users rate their willingness a 9. P1 reported, "Yes, although tedious, the questions made it capable of differentiating junk and important emails."

Furthermore, on a Likert scale between 1 and 5, on average users report their likeliness to use the classifier in the future a 4.6, with 50% of them saying they would use the classifier for personal use-cases, 40% using the classifier for education, and 20% for work. Many users found use cases for BYOC beyond emails. P2 reports: "I could see myself using it in order to give me specific recommendations on things to eat, watch, or do and also to sort emails." P3 stated a more specific use-case for themselves: "I could use a personalized classifier to classify homework assignments in college to figure out which ones should be prioritized."

Beyond the higher level questions we asked about BYOC, we also asked users about their experience with specific parts of the experiment. Users noticed how the questions asked by the model reminded them about a lot of the cases that they had forgotten about in their initial class descriptions. P4 states, "Yes, the process was smooth because the questions allowed me to realize further ways to classify the emails that I had initially forgotten." Furthermore, when talking about whether they liked the interactive experience of answering questions, P5 reports: "I did, because they were used to determine what groups certain things went in. The process felt like sorting in real life."

Additionally, all users believe that the model ultimately produced more descriptive and accurate class descriptions than they had written initially. P6 reports, "Yes, the model gave me better descriptions and allowed the classifications to be more accurate to what I wanted."

### 5.3 Comparison With Existing Approaches

For comparison with previous classification literature, we also wish to evaluate BYOC on an existing dataset. We choose the task of academic paper topic classification, in which the abstract of an academic paper is classified into one of a fixed set of topics. We choose this task because it requires domain-specific knowledge, especially around the specific usage of each class, impairing the zero-shot ability of LLMs. The task also involves longer text, which limits the number of few-shot examples that an LLM can process.

**Dataset** We use the WOS-5736 dataset by Kowsari et al. (2017). The dataset contains eleven classes of research paper abstract topics, with three parent classes: biochemistry, electrical engineering, and psychology.

For each sub-class, we selected 3 abstracts at random belonging to that class, and annotated them with BYOC, answering 3 questions each.

As we are not experts in the fields covered by the dataset, we consulted online sources, including Google and ChatGPT, to provide the initial class descriptions and answer the model's generated questions. Answering each question took us around 3-5 minutes on average. We assume that that end-users will be classifying text that they are familiar with. It is reasonable to assume that domain experts, with access to the same resources and knowledge bases, would have better answers with a less amount of effort to annotate the data. Note that incorrect answers can only impair accuracy; our results is thus a lower-bound of what can be achieved by a domain expert. An example of question and answer is in Fig. 2.

The state-of-the-art approach on this dataset leverages a hierarchical model, classifying first the text into one of the parent classes, and then into one of the sub-classes. We use the same approach for this experiment. As the abstracts are short, no summarization was necessary for this experiment.

The questions, answers, and generated class descriptions for the experiments are available online[2].

**Results** Table 3 compares BYOC against the state of the art. Our best result improves over the few-shot approach by 9%, while consuming an average of 1,167 fewer tokens per sample.

When compared to the state-of-the-art approach for this dataset (Javeed, 2023), which used the full training set of 4,588 examples, our model is only 20% worse. If we take the state-of-the-art result as an upper bound over this dataset, BYOC achieves 79.2% of the maximum possible accuracy with less than 1% of their training set.

---

[2] https://github.com/bardeenai/byoc-dataset

| Approach | # Tokens | Accuracy |
|----------|----------|----------|
| Zero-Shot | **631** | 58.7% |
| + Few-Shot | 3,873 | 65.9% |
| Non-hierarchical | 2,276 | 67.0% |
| BYOC (GPT-3.5) | 2,706 | **74.8%** |
| BYOC (GPT-4) | 4,656 | 63.2% |
| HDLTex | n/a | 90.9% |
| ConvexTM | n/a | 92.42% |
| Hawk | n/a | 94.45% |

Table 3: Accuracy and token counts on the Web of Science test set. Results for HDLTex are as reported by Kowsari et al. (2017), ConvexTM by Bhattarai et al. (2022) and Hawk by Javeed (2023). Our approaches use a few-shot training set while the others use the full training set.

We also compare the use GPT-3.5 during construction of class descriptions, as in previous experiments, with the use of GPT4 throughout. Note that GPT4 was in all cases used for the classification after obtaining the class descriptions. We observe that class descriptions obtained from GPT4 achieve lower accuracy. From our observations we see that GPT4 tends to generate class descriptions that are longer and more specific to the given training examples, leading to a form of overfitting.

Finally, we also evaluated BYOC in a non-hierarchical setting where it attempts to directly predict one of 11 classes, using the same class descriptions. We found it materially worse than the hierarchical case, with an accuracy of 67% on the test set. This suggests that there is a limit to the number of classes that an LLM can predict using class descriptions, and the hierarchical approach is to be preferred.

## 6 Conclusion

This paper presented a novel approach to few-shot text classification using large language models (LLM). By having the user cooperate with an LLM we were able to build highly accurate classifiers while annotating a very minimal amount of data.

For many practical applications of classification, our method eliminates the need for large annotated corpora. We have achieved high accuracy classifiers that are competitive with models trained with orders of magnitude larger datasets.

We also allow end-users to create their own classifiers without expecting them to be proficient in prompt engineering. Through an interactive pro-

cess of question-answering and minimal annotation we are able to make LLM capture the essential features of each class, enabling a precise classification.

Our experiments highlights the importance of both personalizing and user-interaction with the LLM. First, we show that users do not know how to instruct LLMs. Second, we demonstrate the need for personalization. A fine-tuned model trained on all users at once is outperformed by BYOC by 13%, despite outperforming all few-shots baselines. This suggests that there is no one-size-fits-all model that can be applied to every-user. Our study involving real participants also confirms the usability and effectiveness of our approach. End-users are able to build classifiers for themselves exhibiting an average accuracy of 90%, rating their experience 8.6 out of 10.

Overall, our methodology enables users to leverage the power of LLMs for their specific needs, opening up new possibilities for practical applications of text classification in various domains.

This work is available as a commercial product in the Bardeen web automation platform[3], where BYOC enables end-users to classify several forms of data. Popular use-cases include sales lead, inbox classification, and support request categorization.

## Limitations and Future Work

We have identified the following limitations of our approach:

- BYOC uses the LLM as a text-based interface. This is a design decision driven by the availability of API-based LLMs, but it might limit the maximum achievable accuracy, compared to fine-tuning the model or using a method that has access to LLM embeddings or logits. Additionally, using an LLM without fine-tuning exposes a risk of hallucination, and it is possible that the user is misled during training.

- For long inputs it is possible that, due to the summarization step, relevant criteria are omitted from the input of the model, which would lead to a wrong classification. While our summarization technique is preferable over truncating the input, future work should explore additional summarization methods or alternatively encodings of the input.

- Practically speaking, the number of samples that the user is willing to annotate is usually

[3]https://bardeen.ai

very low, and it is therefore likely that rare features of the text would not be accounted for in the classification prompt because they would not be seen by the model during training and not brought up by the user. We envision future work will investigate ways to improve the classifier while in use, perhaps by detecting if the text does not seem to conform to any of the classes according to the given descriptions.

Additionally, in order to keep the experiment consistent across users, we only evaluate in the personalized setting with one binary classification task, namely classifying important and unimportant emails. Future work should investigate how the approach generalizes to different tasks, whether there are tasks that benefit more or less from personalization, and whether the number and detail of classes affect the quality.

## Ethics Statement

We see two main risks of personalized classification in the real world. First, it is possible that users, who are not machine learning experts, are unable to correctly estimate the quality of the classifier, and rely on it always being correct. This is particularly problematic if the validation set used to assess the model quality is small and not representative. As discussed in the limitations sections, to mitigate this issue, we expect that real-world deployment of BYOC will include a mechanism to detect when the model should abstain from making a classification.

The second concern is the privacy risk from the use of third-party APIs on sensitive personal data. OpenAI does not store or train on data processed through their API. In the future, we expect particularly sensitive applications will use open-source models such as LLaMa (Touvron et al., 2023), running on a device of the user's choice.

Specifically regarding our user study, the study was conducted with willing adult participants that expressed informed consent in writing, recruited through direct connections. Participants were compensated for taking part in the study. The email dataset will not be released. Examples provided in the paper are from the authors and not the study participants.

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

# A Question Generation Example

**Text**

The significant co-occurrence between men's violence against female partners and child abuse and neglect is well documented. It is less clear how child safety should be managed in family violence research with their mothers. This issue is salient to isafe, a New Zealand-based Internet intervention study testing improvement in safety decisions and mental health outcomes for women experiencing intimate partner violence. This article discusses the legislative, professional, and ethical considerations that contribute to the development of the child safety protocols and discusses the development of the isafe protocol. Hypothetical scenarios of the application of the isafe protocol are used to illuminate the issues and provide a basis for future discussion.

**Questions**

1. *Question*: Are there any specific keywords or phrases in the text that are commonly associated with any of the classes?
   *Answer*: abuse; children; ethics; responsible conduct of research

2. *Question*: In the context of this text, what criteria would differentiate the child abuse class from the depression class?
   *Answer*: The criteria differentiating the child abuse class from the depression class include the nature of the behavior (physical, emotional, or sexual abuse towards a child), the perpetrator-victim relationship (parent or caregiver abusing the child), the focus on child safety and well-being, and legal obligations surrounding child protection.

3. *Question*: Are there any specific ethical considerations mentioned in the text that could help us better understand the criteria for the child abuse class?
   *Answer*: The specific ethical considerations mentioned in the text are not provided, making it difficult to precisely identify criteria for the child abuse class. However, based on the text's context, it can be inferred that ethical considerations could include ensuring the safety and well-being of the child, protecting their rights, maintaining confidentiality, adhering to professional standards, and navigating the complexities of intervening in cases of child abuse. These considerations likely inform the development of child safety protocols and guide decision-making processes in the iSafe study.

Figure 2: Example of questions and answers generated on one abstract from the Web of Science dataset (Kowsari et al., 2017).

## B  Prompts Used for Experiments

### B.1  Question Generation

System Instructions:

You are taking part in an interactive study where the goal is to build a personalizable classifier for a user. This means strictly following the user's class descriptions and criteria for classification. If a user doesn't cover a certain case, it means trying to find patterns in the user's thoughts or criteria to make a best guess.

The goal of this stage in the process of building a classifier is to annotate some training data, so we can refine the user's class description and better understand their criteria. The way we choose to annotate our training data is by asking user's questions about the text to better help us understand the classification task at large.

This doesn't mean asking questions about what the current text means or the specific context of the text, as answering these questions won't always allow us to generalize the answer to other examples.

The questions that you ask must be ones that if answered, could help us broaden and improve the scope of the user's class descriptions. The questions should relate to the class names and descriptions provided, and should even ask clarifying questions about the class descriptions if needed. The questions should not be vague or obvious. They should be questions that only the user who is trying to perform classification would know, not questions with answers that are obvious to anyone.

Some other questions may include how might you differentiate between classes, if you are deciding between two classes for this specific text.

You could also ask if there are any specific keywords or phrases that are commonly associated with a certain class. Also, do not just directly ask the user what class the text belongs to – this is not the goal of this stage. Don't just ask these question, since these are just examples of questions you could ask.

In the prompt, we also display questions that were previously asked by the model, followed by the user's responses to those questions. If needed, your questions should build upon previously asked questions and answers, so you can better understand the user's intent.

Additionally, along with the question you ask, also explain why you asked the given question and what you hope to understand from the user by asking the question.
Your response should be formatted like:

Thoughts: <thoughts>
Question: <question>
Explanation: <explanation>

User Instructions

For this classification task, the classes that the user chose were: <class_1>, <class_2>, ....
<class_n> name. Here is a description of the task at hand (why the user wants to build this classification task): <classification_task_description>

Each class also has a corresponding class description, which describes the criteria the text must follow to belong to a given class. Here are the class descriptions of each class.

<class_1_name>: <class_1_description>
<class_2_name>: <class_2_description>
..
<class_n_name>: <class_n_description>

Here is the current text that we are annotating:
--- Start of text ---
${text}
--- End of text ---

Based on this text and the classes we are trying to classify the text under, we generated the following questions and answers about the text to help us classify it. There might be no questions generated yet.

```
<question_1>
<answer_1>

<question_2>
<answer_2>

<question_m-1>
<answer_m-1>

We want to ask another question about the text to help us improve or broaden the scope of the class
descriptions, as per the instructions provided. The question could also be a follow up question to
the previous questions. What should it be and why?;
```

## B.2   Few-shot Classification (During Interactive Annotation)

System Instructions:

```
You are taking part in an interactive study where the goal is to build a personalizable classifier
for a user. More specifically, you need to create a classifier that classifies the text of a user
into classes of their own choosing. This means strictly following the user's class descriptions and
criteria for classification. If a user doesn't cover a certain case, it means trying to find
patterns in the user's thoughts / criteria to make a best guess.

The goal of this stage in the process is to classify training data, so we can understand if the
current class descriptions are covering enough use-cases for them to be generalized and deployed for
all texts. In this prompt, we are focusing on a single text and trying to classify it into one of
the classes the user has inputted.

Additionally, for your reference, we have also asked the user a series of questions to help us
classify this text. Use those questions in order to make your classification.

If you are not provided any information or questions, then make your best guess as to what class the
text belongs to.

Furthermore, beyond just the classification, also explain your thoughts as you decide what class the
text belongs to. After making the classification, then reflect on why you chose that class. Your
response should be formatted like:

Thoughts: <thoughts>
Class: <class>
Reflection: <reflection>
```

User Instructions:

```
For this classification task, the classes that the user chose were: <class_1>, <class_2>, ....
<class_n> name. Here is a description of the task at hand (why the user wants to build this
classification task): <classification_task_description>

Each class also has a corresponding class description, which describes the criteria the text must
follow to belong to a given class. Here are the class descriptions of each class.

<class_1_name>: <class_1_description>
<class_2_name>: <class_2_description>
..
<class_n_name>: <class_n_description>

Here is the current text that we are annotating:
--- Start of text ---
${text}
--- End of text ---

Based on this text and the classes we are trying to classify the text under, we generated the
following questions and answers about the text to help us classify it. There might be no questions
generated yet.

<question_1>
<answer_1>
```

```
<question_2>
<answer_2>

<question_m>
<answer_m>

Now, given this information, classify the text above - make sure to incorporate your thoughts and a
reflection as well.';
```

## B.3    Refining Class Descriptions

System Instructions:

```
You are taking part in an interactive study where the goal is to build a personalizable classifier
for a user. More specifically, you need to create a classifier that classifies the text of a user
into classes of their own choosing. This means strictly following the user's class descriptions and
criteria for classification. If a user doesn't cover a certain case, it means trying to find
patterns in the user's thoughts / criteria to make a best guess.

The goal of this stage in the process is to update the class descriptions that the user has
inputted, based on the training data we are annotating. In this prompt, we are focusing on a single
text, which the user has answered questions about and explained to us why that example belongs in a
given class.

Using this information, and the correct class of the text, we need to update the class description
of the correct class, to broaden it's scope to contain more examples/possibilities.
Furthermore, beyond just the classification, also explain your thoughts as you decide what class the
text belongs to. After making the classification, then reflect on why you chose that class. Your
response should be formatted like:

Thoughts: <thoughts>
Description: <updated_class_description_for_class>
Reason: <reason_why_you_updated_the_class_description>
```

User Instructions:

```
For this classification task, the classes that the user chose were: <class_1>, <class_2>, ....
<class_n> name. Here is a description of the task at hand (why the user wants to build this
classification task): <classification_task_description>

Each class also has a corresponding class description, which describes the criteria the text must
follow to belong to a given class. Here are the class descriptions of each class.

<class_1_name>: <class_1_description>
<class_2_name>: <class_2_description>
..
<class_n_name>: <class_n_description>

Here is the current text that we are annotating:
--- Start of text ---
${text}
--- End of text ---

Based on this text and the classes we are trying to classify the text under, we generated the
following questions and answers about the text to help us classify it. There might be no questions
generated yet.

<question_1>
<answer_1>

<question_2>
<answer_2>

<question_m>
<answer_m>

Using these questions and answers, we classified the text. Our initial classification was
<model_prediction>. The actual class of the text is <correct_class> The user explained the reason
that this was the correct class was this explanation: <user_explanation>
```

We want to update the description of the class <class_to_be_updated> to be more accurate based on this new information. Without changing or altering the meaning in the current description, what is a better class description for the class: <class_to_be_updated>

If there is nothing to add to the current class description from this example, then just copy the current class description word for word.

## B.4 Summarization

System Instructions:

I had a long thread of text and I wanted to take the summary, so I split it into smaller parts. We will provide the summary of the first <i-1> parts of the essay and the summary of the  part.

Use these summaries as context to provide a summary of the entire essay, Don't make the length of the summary of this one part of the text equal to length of the the summary of multiple parts of text.

Make the summary of the new thread proportional to how many parts of the text have been summarized so far.

For example, if the summary of about 5 parts of the text are 400 words, the length of the summary of the sixth part should be around 80 words.

Remember, the focus of the summary should also be on: <classification_task_description>;

User Instructions:

Summarize the following text with a focus on preserving context. Format the reponse with style: free-text. Summarize with the context: <classification_task_description> and with a focus: we want to eventually perform classification on this text.

Summary of First <i-1> parts: <summary_of_previous_sections>
Part : <current part>