# OpenReview forum: "BYOC: Personalized Few-Shot Classification with Co-Authored Class Descriptions"
_EMNLP/2023/Conference — EMNLP 2023 Findings_

### Official Review · Reviewer_cNMD · 2023-08-04

**Soundness:** 4

**Excitement:**

4: Strong: This paper deepens the understanding of some phenomenon or lowers the barriers to an existing research direction.

**Missing References:**

HDLTex is technically not the SOTA for the WOS-5736 dataset, there are some newer works - ConvTextTM (https://aclanthology.org/2022.lrec-1.401), Hawk (https://arxiv.org/abs/2301.06057).


**Paper Topic And Main Contributions:**

In this work, the authors propose a new approach for using LLMs to perform text classification with a human in the loop, which is oriented towards solving a personalized classification problem given a limited amount of feedback from the user. The basic approach is that the phase of interaction between the LLM and the user is used to take the description of the the target classes initially given by the user, and refine them based on a set of few-shot examples and some back-and-forth between the LLM and the user. Then, at inference, the refined class descriptions are given as input to an LLM in order to decide how to classify a new input example. The authors conducted a test study with users that use this interactive feedback mechanism to construct a personalized binary classifier for each user. The authors show that this approach outperformed other baselines for incorporating the different kinds of inputs given by the users for classification by an LLM. In addition, they report the performance of this approach over an academic classification dataset.

**Questions For The Authors:**

A. You state that "The model sees both the correct and predicted labels, which allows it to weigh how important is the new example" (line 385). Is there a basis for assuming that this is indeed what the LLM does with this information?


**Reasons To Accept:**

- The paper deals with some very important and under-explored real world problems, namely those of building personalized classifiers and of utilizing interactive feedback with a human in the loop and different kinds of user input, and not just standard application of an algorithm to a pre-labeled academic classification dataset.
- The concept that the ultimate goal of interaction with the user is to arrive at refined descriptions of the target classes, rather than a labeled (or pseudo-labeled) set of examples, and that this would better serve the classifier, is an interesting one, and may lead the community in novel directions.
- The overall methodology looks sound.
- The work is well-written and easy to follow.

**Reasons To Reject:**

- I would have liked to see more discussion of the way the prompts were engineered, given that these prompts look extremely complex. First, it is not so clear that the LLM (even GPT-4) is truly able to comprehend such long and complex instructions, nor that all components of this instruction are actually necessary. And second, I wonder what was the process by which the authors arrived at this particular set of instructions.
- Presumably this approach would not scale easily to a scenario with many classes and/or with very fine-grained class distinctions — in terms of both computation, annotation effort, and performance — mainly since a larger amount of questions and increasingly complex user answers would be required to construct a good description for each class. In such a scenario, it is likely that given the same amount of effort by the domain expert, existing approaches for few-shot fine-tuning would ultimately reach superior results.
On this note, while the results for WOC are encouraging, obtaining multiple user answers with a quality similar to the example in App. A is no easy feat. I will say that while I agree with the claim that the results can be considered a lower bound on what *can* be achieved by a domain expert, this does not necessarily imply that it is a lower bound on what *will* be achieved by a domain expert in a realistic scenario, at least without devoting a very significant amount of time both on the part of the domain expert and on properly guiding the expert on the importance and expected form of the provided answers. Given the significant overhead, the question would then turn to whether a given amount of effort would be better spent on complex feedback to the LLM or on more traditional labeling of a larger set of training examples for fine-tuning. This is of course an open question.
- While I completely agree on the importance of being able to build personalized classifiers, I am not so convinced by the text in lines 544-559. The model, methodology, and type of input data are all very different for BYOC and the fine-tuned setting, it is difficult to conclude from this experiment that the performance difference stems specifically from personalization.


**Reproducibility:**

3: Could reproduce the results with some difficulty. The settings of parameters are underspecified or subjectively determined; the training/evaluation data are not widely available.

**Reviewer Confidence:**

4: Quite sure. I tried to check the important points carefully. It's unlikely, though conceivable, that I missed something that should affect my ratings.

**Typos Grammar Style And Presentation Improvements:**

- I would suggest changing the notation of the "classifier purpose", just because _p_ is strongly associated with the context of probabilities whereas here the meaning is of a class description.

---

> ### Author Rebuttal · Authors · 2023-08-28
>
> Thank you for your thoughtful and thorough review.
>
> **The Design of Our Prompts**
> We gradually improved a basic prompt by adding instructions, observing performance on large amounts of data, and author interaction. This resulted in the prompt presented in our paper.
>
> Specifically for the question-generation prompt, we observed that the LLM generated similar questions for each example. As a result, we added an extra section in our prompt in which we instructed the model to not only ask very general questions. Similarly, in the same stage, we observed that the model occasionally asked a redundant question on further examples, such as "Are meetings important?" on multiple meeting emails. We added a section in the prompt to overcome that. Overall, the question-generation prompt was optimized to broaden the range and scope of the user's class descriptions.
>
> For the class description refinement prompt, specifically in response to question A, we observed that updating the class description for correct and incorrect predictions is necessary. We found that we could only achieve that by providing the model with both the incorrect prediction and the correct label.
>
> Finally, for the prompts used to classify the test set, we manually tuned using the acquired validation data from the personalized experiments, following standard practice. We will include this discussion of our prompt design in the final revision of our paper, taking advantage of the extra page. We also discuss our prompt designs in the response to Reviewer ixK3.
>
> **The Scalability of Our Approach**
> We understand the concern about the scalability of our approach. In the comparison on Web of Science, which uses a hierarchical approach to reduce the number of classes, we also evaluated BYOC in a non-hierarchical setting where it attempts to predict one of 11 classes directly. We found it materially worse than the hierarchical case, with an accuracy of 67% on the test set. We will include a discussion of this result in the final revision of this paper.
>
> Furthermore, assuming that a fine-tuned specialized model will perform similarly to BYOC is not unreasonable. However, we cannot expect non-technical end users to be able to fine-tune a model for every user case. We can also assume that an end-user would not go through the process of annotating large datasets to fine-tune a model for their classification tasks. Therefore, the alternative is to use a one-size-fits-all model, which we proved to have lower accuracy than the proposed approach.
>
> **Analyzing the User's Ability to Effectively Answer the Model's Questions**
> Regarding the user's ability to answer the model's questions effectively, while we did not evaluate with experts on WOS, our personalized experiment should be a good proxy of what is achievable, as everyone is an expert of their inbox. Our results outperform existing few-shot fine-tuning by a large margin. On WOS, our answers were provided by non-experts and, on average, required 3-5 minutes each. As such, it is reasonable to assume that experts with access to the same tools and knowledge bases would have similar or better answers. The amount of effort would still be lower than annotating enough training data. Conducting a user study is one possibility for further analysis of this dataset.
>
> **The 'Importance of Personalization' Experiment**
> Our "Importance of Personalization" experiment aims to compare a model trained on one user's data with a model trained on all users' data. Unfortunately, a few-shot method that applies to only one user cannot be used with all users because the prompt would be far beyond the context window of any LLM. Hence, we need to compare a few-shot method to a fine-tuned method. The choice of fine-tuning BERT, rather than a larger model, was then dictated by computational constraints. Future work could evaluate larger fine-tuned models such as LLaMA.
>
> **The Missing Citation and Style**
> Finally, regarding the citations we missed (ConvTextTM and Hawk) and changing the notation of "classifier purpose", we thank you for noting these down, and we promise to include them in the revision of the paper.

---

### Official Review · Reviewer_ixK3 · 2023-08-06

**Soundness:** 3

**Excitement:**

3: Ambivalent: It has merits (e.g., it reports state-of-the-art results, the idea is nice), but there are key weaknesses (e.g., it describes incremental work), and it can significantly benefit from another round of revision. However, I won't object to accepting it if my co-reviewers champion it.

**Missing References:**

None

**Paper Topic And Main Contributions:**

The paper shows how the effectiveness of few-shot training examples can be improved through the interaction with the foundation LLM, in a manner similar to active learning approach.

**Questions For The Authors:**

None

**Reasons To Accept:**

It is an interesting idea and timely topic due to popularity of LLMs and not yet their properties well understood.

**Reasons To Reject:**

The result is not that surprizing and the details are not yet presented in the best way so the paper may benefit from revisions. The findings can be very sensitive to specific details like prompts used. But the sensitivity to those and ablations have not been run. The specific choice of prompts has not been justified, e.g. the LLM is expected to operate with the concepts “study” and “essay”. It would be interesting to probe that directly how well it can do that .

The setup is a bit artificially since if we assume that GPT4 is the best model around to assess the results, then why we don’t use it directly for classification and all the refinements as well? Smaller GPT3 is used instead for those.


**Reproducibility:**

4: Could mostly reproduce the results, but there may be some variation because of sample variance or minor variations in their interpretation of the protocol or method.

**Reviewer Confidence:**

4: Quite sure. I tried to check the important points carefully. It's unlikely, though conceivable, that I missed something that should affect my ratings.

**Typos Grammar Style And Presentation Improvements:**

Examples in Figure 1 is a bit simplistic since only shows how “appointment” topic is added to the list of non-spams.

Section 4 formal descriptions are hard to follow without specific examples.  Same applies to section 3.1.

More diversity of the included transcripts would help: e.g. the instruction “You are taking part in an interactive study” repeated several times making up a substantial portion of the presented transcript.

Some system-aspect details can be skipped if they are not important for the experiment, e.g. “she can share it with others, like family  members with similar email needs” or “the user selects the source of data”

---

> ### Author Rebuttal · Authors · 2023-08-28
>
> Thank you for your thoughtful and thorough review.
>
> **The Novelty and Expectedness of the Paper**
> Our paper demonstrates that using an LLM with zero-shot class descriptions leads to poor performance. This is surprising given the high performance of models like GPT4 in many tasks. We find that significant effort is required for prompt tuning, which end-users can only achieve in an interactive, guided setting. Please also see the response to Reviewer JyXx for further elaboration on this point.
>
> **The Sensitivity of our Results to the Prompt**
> Regarding our system design and sensitivity to details, we first optimized our prompts and hyperparameters through the author's interaction with the system on internal sample data by analyzing the model's output. We used both chain-of-thought and self-reflection to improve our prompts. Since our approach is interactive, changing a design choice that affects the user study is challenging and requires multiple user studies, likely with multiple user groups and larger sample sizes, to achieve significance. On the other hand, for the final experiment on both personalized and non-personalized data, the prompts were optimized on the validation data, as is standard practice. We conducted ablation studies (Table 2) of each aspect of our approach.
>
> Specifically, regarding the instruction about taking part in an interactive study, we observed that without that instruction, the model would generate questions and outputs more tied to the specific input given (email or paper abstract). In contrast, with the given instructions, we qualitatively observed that the model behaved as if it was part of a longer context.
>
> We will include further discussion of how we designed our prompts in the final revision of our paper, taking advantage of the additional page. We also discuss our prompt designs in the response to Reviewer cNMD.
>
> We aim to build a practical classification approach and evaluate its usability with real users. We included the additional system details to emphasize the overall usability. These details are relevant to contextualize this work and give validity to the survey portion of our user study.
>
> **Our Choice to use GPT-3.5**
> Regarding using GPT-3.5, our choice was due to speed and cost, especially in the interactive setting where speed can be a factor for usability. We will include an offline experiment that exclusively uses GPT-4 in the final version of the paper.
>
> We thank the reviewer for the suggested improvements to the style and presentation of the paper and will make sure to include those in the final version.

---

### Official Review · Reviewer_JyXx · 2023-08-11

**Soundness:** 3

**Excitement:**

3: Ambivalent: It has merits (e.g., it reports state-of-the-art results, the idea is nice), but there are key weaknesses (e.g., it describes incremental work), and it can significantly benefit from another round of revision. However, I won't object to accepting it if my co-reviewers champion it.

**Paper Topic And Main Contributions:**

The paper proposes a few-shot text classification approach using an LLM. Rather than few-shot examples, the LLM is prompted with descriptions of the salient features of each class. These descriptions are coauthored by the user and the LLM interactively: while the user annotates each few-shot example, the LLM asks relevant questions that the user answers. Examples, questions, and answers are summarized to form the classification prompt. The main contribution is to interactively construct class descriptions for classification, by
prompting the LLM to ask relevant questions and enables non-experts to build classifiers with high-accuracy.

**Reasons To Accept:**

The idea is quite user-friendly to enable non-experts to build a high accuracy classifier. The result looks promising showing a 9% improvement over the few-shot state of the art, and reaches 82% of the accuracy of a model trained on a full dataset with only 1% of their training set. The author has also surveyed with 30 participants, who find this approach interpretable and would consider for their use cases.  This could be landed as a very useful application of LLM to build customized classifiers.

**Reasons To Reject:**

I find this paper lacks technical strength or novelty. The application is good but it lacks more novelty in the method. Since it requires user interaction to build the classifier, it makes it hard to exactly reproduce the results and easily compare with other baselines on more datasets. I would recommend the authors to propose a way for others to reproduce the results more easily and how they could leverage the framework for other tasks. And evaluate it beyond one dataset as reported in the paper.

**Reproducibility:**

3: Could reproduce the results with some difficulty. The settings of parameters are underspecified or subjectively determined; the training/evaluation data are not widely available.

**Reviewer Confidence:**

4: Quite sure. I tried to check the important points carefully. It's unlikely, though conceivable, that I missed something that should affect my ratings.

---

> ### Author Rebuttal · Authors · 2023-08-28
>
> Thank you for your thoughtful and thorough review.
>
> **The Importance of User Interaction**
> Our main goal with this paper is to evaluate how we can leverage human interaction to improve the performance of NLP models in the real world, with a specific focus on classification. As Reviewer cNMD also notes, our objective goes beyond the standard application of an algorithm to a pre-labeled academic classification dataset. As such, we structured our experiments around the user study and evaluation in the personalized setting. Indeed, the human aspect makes the paper challenging to reproduce as a benchmark. However, it remains integral to our contribution. We aim to assess whether people can achieve good classification for their own use cases.
>
> Our most surprising result is that the most common way people outside of research interact with LLMs - by providing handcrafted zero-shot instructions - performs very poorly. Users need help to write good instructions, even in a use case they are very familiar with, such as email.
>
> **The Novelty of the Approach**
> We propose a novel interactive approach to refine these zero-shot class descriptions, and we achieve a new state-of-the-art result for few-shot classification with a similar amount of annotation effort - 90% compared to 66% with handcrafted class descriptions (Table 2). As discussed in Section 3.2, we are the first to incorporate interactive and prompt refinement aspects. This combination enables BYOC to use the user’s annotation effort best while preserving the inference-time cost efficiency of a zero-shot-like approach.
>
> Furthermore, unlike prior work, we changed the input of the LLM in a novel way during the annotation process. In previous studies, researchers have used labels and explanations as inputs to their LLM for classification tasks. On the other hand, in BYOC, our inputs to the models are labels, questions/answers, and an explanation. This is far more effective than previous works because:
> - The questions presented to the user are easily understandable to a non-technical end-user, enabling them to provide more relevant and useful answers.
> - The questions increase the LLM’s ‘understanding’ of the classification task since we synthesize these questions into new class descriptions, which enables the class descriptions to capture more relevant information.
> - Using labels, Q&A, and explanations, we pack far more information per sample than previous methods.
> - As a result, we need far less data to annotate, and the annotation process is more relevant and user-friendly.
>
> **Using our Approach in other Tasks**
> While we only evaluate classification in this paper, a similar interactive approach would benefit settings such as recommendation and text generation. Future work should evaluate our approach on those tasks.
>
> **The Reproducibility of the Work**
> For reproducibility and comparison with existing work, we also evaluate on a second dataset, WOS, which is a standard benchmark. We will release all our prompts (included in Appendix A) and all questions and answers used to obtain the class descriptions through BYOC. These are equivalent to the “BYOC dataset” and can be used to train and evaluate future methods.

---

### Meta-Review · Area_Chair_cBeo · 2023-09-19

**Recommendation:** 4

**Metareview:**

This paper presents an approach to learn text classifiers based on interacting with an LLM to annotate a small amount of training examples, with the goal of generating class descriptions for use in a prompting-based approach. They find that their approach outperforms other prompting-based approaches and conduct a user study that indicates positive reception of the approach.

Soundness:

Reviewers give fair "soundness" ratings with 3/3/4 though list several concerns in their written reviews.

Reviewers voice concerns regarding the technical details of the experimental evaluation, namely the focus on only one public benchmark (WOC) and the lack of justification for the specific prompts used. Additionally, since the proposed approach relies on interactivity with humans, this naturally limits reproducibility. The authors however try to mitigate the lack or reproducibility by releasing much information on the conducted study.


Excitement:

Reviewers also give fair "excitement" ratings with 3/3/4.

One limiting factor excitement are presumed difficulties in scaling the proposed approach to many classes, or classes with fine-grained distinctions, in which case a potentially very large amount of interactions would be necessary with the LLM. The authors acknowledge this limitation in their response and will add discussion on this.

The novelty is questioned by some reviewers, but the author response alleviates this by pointing out that their approach outperforms other prompting-based approach significantly, albeit only evaluated on one benchmark dataset and one small proprietary dataset.

---

### Decision · Program_Chairs · 2023-10-07

**Decision:**

Accept-Findings

**Comment:**

This paper presents an approach to learn text classifiers based on interacting with an LLM to annotate a small amount of training examples, with the goal of generating class descriptions for use in a prompting-based approach. They find that their approach outperforms other prompting-based approaches and conduct a user study that indicates positive reception of the approach.

Soundness:

Reviewers give fair "soundness" ratings with 3/3/4 though list several concerns in their written reviews.

Reviewers voice concerns regarding the technical details of the experimental evaluation, namely the focus on only one public benchmark (WOC) and the lack of justification for the specific prompts used. Additionally, since the proposed approach relies on interactivity with humans, this naturally limits reproducibility. The authors however try to mitigate the lack or reproducibility by releasing much information on the conducted study.


Excitement:

Reviewers also give fair "excitement" ratings with 3/3/4.

One limiting factor excitement are presumed difficulties in scaling the proposed approach to many classes, or classes with fine-grained distinctions, in which case a potentially very large amount of interactions would be necessary with the LLM. The authors acknowledge this limitation in their response and will add discussion on this.

The novelty is questioned by some reviewers, but the author response alleviates this by pointing out that their approach outperforms other prompting-based approach significantly, albeit only evaluated on one benchmark dataset and one small proprietary dataset.